# Characterization of a member of the CEACAM protein family as a novel marker of proton pump-rich ionocytes on the zebrafish epidermis

Julien Kowalewski[☉], Théo Paris[☉], Catherine Gonzalez, Etienne Lelièvre, Lina Castaño Valencia, Morgan Boutrois, Camille Augier, Georges Lutfalla*, Laure Yatime[ID]*

Laboratory of Pathogen-Host Interactions (LPHI), UMR5235, University of Montpellier, CNRS, INSERM, Montpellier, France

[☉] These authors contributed equally to this work.
* georges.lutfalla@umontpellier.fr (GL); laure.yatime@inserm.fr (LY)

**Data Availability Statement:** All relevant data are within the manuscript or its Supporting Information files.

## Abstract

In humans, several members of the CEACAM receptor family have been shown to interact with intestinal pathogens in an inflammatory context. While CEACAMs have long been thought to be only present in mammals, recent studies have identified *ceacam* genes in other vertebrates, including teleosts. The function of these related genes remains however largely unknown. To gain insight into the function of CEACAM proteins in fish, we undertook the study of a putative member of the family, CEACAMz1, identified in *Danio rerio*. Sequence analysis of the *ceacamz1* gene product predicted a GPI-anchored extracellular protein containing eleven immunoglobulin domains but revealed no evident orthology with human CEACAMs. Using a combination of RT-PCR analyses and *in situ* hybridization experiments, as well as a fluorescent reporter line, we showed that CEACAMz1 is first expressed in discrete cells on the ventral skin of zebrafish larvae and later on in the developing gills. This distribution remains constant until juvenile stage is reached, at which point CEACAMz1 is almost exclusively expressed in gills. We further observed that at late larval stages, CEACAMz1-expressing cells mostly localize on the afferent side of the branchial filaments and possibly in the inter-lamellar space. Using immunolabelling and 3D-reconstructions, we showed that CEACAMz1 is expressed in cells from the uppermost layer of skin epidermis. These cells are embedded within the keratinocytes pavement and we unambiguously identified them as proton-pump rich ionocytes (HR cells). As the expression of *ceacamz1* is turned on concomitantly to that of other known markers of HR cells, we propose that *ceacamz1* may serve as a novel marker of mature HR cells from the zebrafish epidermis.

**Funding:** This work was supported by a grant from the European Community's H2020 Program Marie-Curie Innovative Training Network ImageInLife: Grant Agreement nr. 721537, awarded to GL (http://imageinlife.eu/). The funders had no role in study design, data collection and analysis, decision to publish, or preparation of the manuscript.

**Competing interests:** The authors have declared that no competing interests exist.

## Introduction

Carcinoembryonic antigen-related cell adhesion molecules (CEACAMs) are cell-surface glycoproteins that belong to the immunoglobulin (Ig) superfamily [1,2]. They are part of the larger CEA group of proteins which also contains pregnancy-specific glycoproteins (PSGs) [1–3]. CEACAMs are only found in higher eukaryotes. For a long time, they were believed to be mammal-specific but more recent studies have identified orthologues in amphibians and teleosts, suggesting that these proteins are widespread among vertebrates [4,5]. CEACAM5, the prototype of the family initially termed CEA, was discovered in 1965 by Gold and Freedman in carcinomas from human colon, as well as in some digestive fetal tissues [6,7]. It was originally thought to be absent from adult normal tissues, therefore acting as a specific antigen for tumor cells in humans, which led to the generic name for the whole CEA family. In the early 1970s, the discovery of both CEACAM5 and other members of the family in non-cancerous tissues allowed to propose that CEACAM proteins have broader functions in the human body besides their role as tumorigenic factors [8–11]. To date, twelve CEACAM genes, all clustered on the q13.2 region of chromosome 19, have been identified in humans [2]. They encode twelve distinct membrane-associated glycoproteins, several of which being expressed as multiple isoforms (Fig 1) [12]. CEACAMs are mostly expressed on epithelial cells, as well as endothelial and myeloid cells (granulocytes, monocytes) [2]. Tissue distribution varies from one protein to the other but most CEACAMs are highly expressed in the gastrointestinal tract or in the bone marrow [2,13]. CEACAM1, CEACAM5 and CEACAM6 have the broadest expression pattern and are present in various human tissues, including skin, lung, liver, kidney or reproductive organs depending on the CEACAM protein considered [13–15].

The extracellular domain of all CEACAMs is composed of a single N-terminal Ig-domain of variable type (Ig-V) followed by none or up to six constant Ig-domains of C2 type (Ig-C2) [12,16]. It is anchored in the plasma membrane either through glycosylphosphatidylinositol (GPI) or via a single-pass transmembrane helix followed by a short cytoplasmic tail containing immunoreceptor tyrosine-based activation/inhibition motifs (ITAM/ITIM) (Fig 1) [12]. The sole exceptions to this general description are CEACAM20, which only bears a partial Ig-V N-terminal domain, and CEACAM16, a secreted protein which contains two Ig-V domains on both its N- and C-termini [12,16]. All CEACAMs are heavily glycosylated, mainly on asparagine residues, and the carbohydrate structures can constitute up to 60% of their total molecular weight [17–19]. CEACAMs primary function resides in cell adhesion and intercellular communication [20,21]. As their extracellular domain is exclusively composed of Ig domains, CEACAMs tend to self-associate through homophilic interactions or to engage in heterophilic contacts with other Ig-containing cell-surface proteins [21–23]. In the tumor microenvironment, these properties are thought to shape the communication between cancer, stromal and immune cells, thereby influencing numerous cellular processes during cancer progression such as cell proliferation, motility, angiogenesis, metastasis or immune responses [2,16]. In both normal and cancer tissues, intercellular contacts mediated by epithelial members of the CEACAM family have also been implicated in apoptosis [24], cell migration [25,26], vascular remodeling [27] and morphogenesis. Interestingly, a number of CEACAMs present in the digestive tract are also used as anchoring points by several intestinal pathogens including pathogenic *Escherichia coli* [28,29], *Moraxella catarrhalis* [30,31], *Haemophilus influenzae* [31], *Helicobacter pylori* [32] and several neisserial strains [33]. These interactions have been linked to intestinal inflammation and have deleterious effects in patients suffering from inflammatory bowel diseases [34]. Clearly, the presence of CEACAM molecules on the epithelium from various organs, and their uniform composition with Ig-domains prompt to engage in protein-protein interactions, make them attractive platforms for extracellular microorganisms, and even

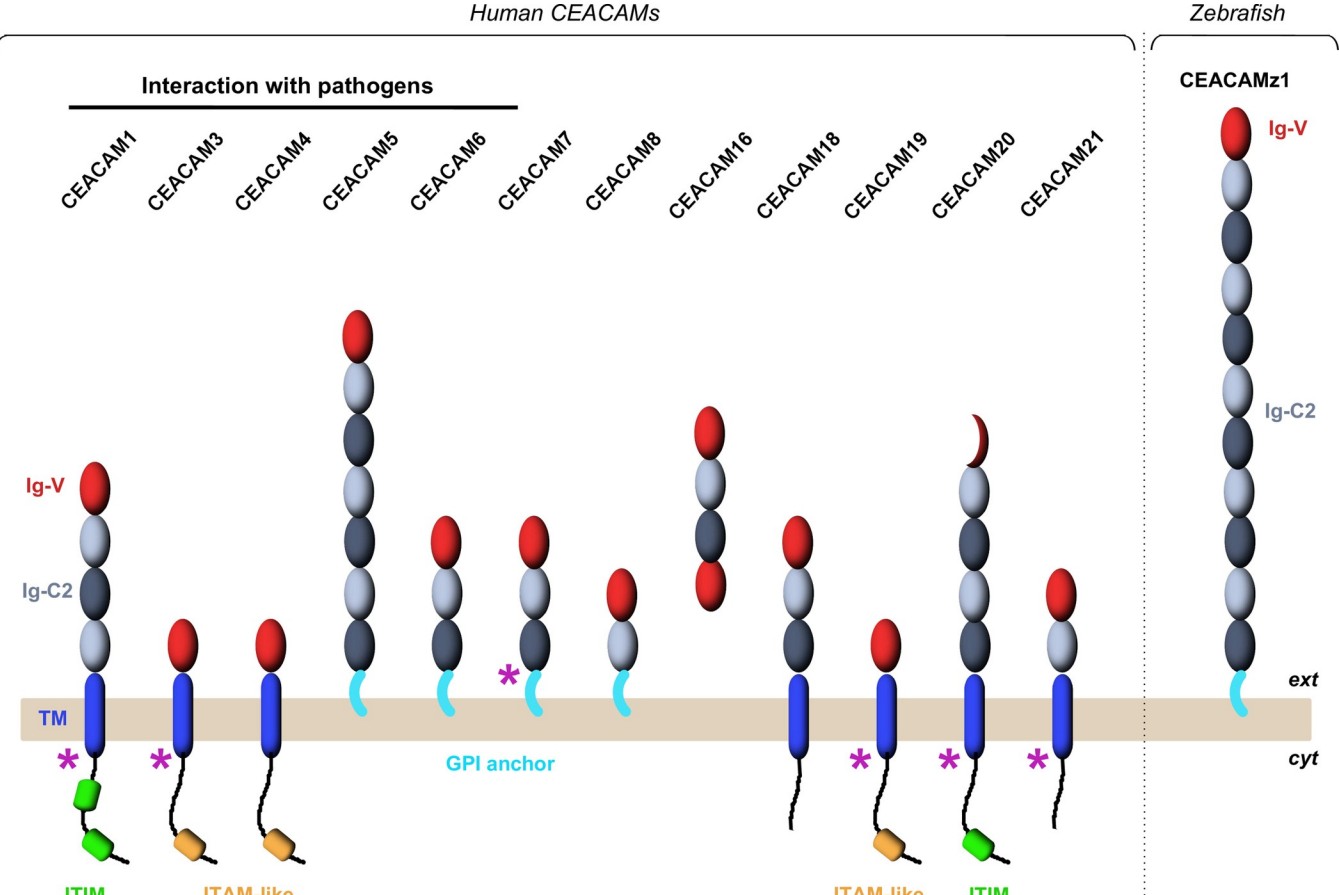

**Fig 1. Schematic representation of the domain organization of the 12 human CEACAM receptors and of zebrafish CEACAMz1.** Immunoglobulin domains composing the CEACAM ectodomains are displayed as oval spheres (in red for Ig-V domains; in light or dark grey for A-type or B-type Ig-C2 domains, respectively). Single-pass transmembrane helices are represented by blue, rounded rectangles. GPI anchors are shown in cyan. Cytoplasmic ITIM (Immunoreceptor Tyrosine-based Inhibitory Motif) and ITAM-like (Immunoreceptor Tyrosine-based Activation Motif) motifs are highlighted in green and orange, respectively. Purple arrows indicate the CEACAMs for which several isoforms have been characterized.

more so for human pathogens. As CEACAM orthologues have been found in more than twenty distinct vertebrate species representing various taxonomic classes [4,5], it is tempting to speculate that this role as a platform for host-pathogen interactions may be conserved among higher vertebrates. However, the function of these putative CEACAM orthologues has not been studied so far in non-mammal species.

Zebrafish is a widely used animal model originally set up by developmental biologists and geneticists. Over the last decades, it has emerged as a powerful tool to study human pathophysiology [35,36]. A a broad repertoire of disease models has been generated in this organism, either through genetic manipulation or via exposure to causing agents, including infection models for various human pathogens such as pathogenic *E. coli* [37], mycobacteria [38,39], salmonella [40] or *Pseudomonas aeruginosa* [41]. The close homology between zebrafish and human innate/adaptive immune systems, as well as the classical advantages inherent to this model (ease of manipulation, genetic amenability, fast generation and abundant progeny, transparency of the larvae) make it a remarkable model to study immune responses elicited by host-pathogen interactions [42]. Zebrafish therefore appears as an attractive alternative to mouse models to deeper study the pathophysiological consequences of CEACAM-pathogen

interactions on intestinal homeostasis. However, no functional analysis of the putative CEA-CAM orthologues identified in *Danio rerio* has been conducted so far. In fact, only one zebra-fish gene is currently annotated in the genomic database as a *ceacam* gene and further characterization still awaits. Here we present a detailed sequence analysis of this gene and its product, that we renamed CEACAMz1. Using RT-PCRs, *in situ* hybridization and fluorescent reporter lines, we further show that at early developmental stages, CEACAMz1 is mainly expressed on the zebrafish skin epidermis, in cells intercalated between keratinocytes. Co-localization experiments using specific markers for the different types of epidermal cells reveal that CEACAMz1 is expressed in ionocytes enriched in $H^+$-ATPase, which are predominantly found in the gill epidermis at adult stage. These findings lead us to propose that CEACAMz1 may serve as a novel marker for proton pump-rich cells on the fish epidermis and may possibly be involved in communication with the extracellular environment.

## Materials and methods

### Ethics statement

All animal experiments described in the present study were conducted at the University of Montpellier according to the European Union guidelines for handling of laboratory animals (http://ec.europa.eu/environment/chemicals/lab_animals/home_en.htm). All experiments were approved by the Direction Sanitaire et Vétérinaire de l'Hérault and by the Comité d'Ethique pour l'Expérimentation Animale under the references CEEA-LR-B4-172-37 and APA-FIS#5737-2016061511212601 v3. Animal studies reported in this paper are in compliance with the ARRIVE guidelines 2.0 [43].

### Zebrafish lines and maintenance

Fish and embryo maintenance, staging and husbandry were as described previously [44]. Experiments were performed using the AB and golden zebrafish strains (ZIRC) or the follow-ing transgenic lines: *Tg(ceacamz1:mCherry-F) ump9Tg* (this study), *Tg(il1b:eGFP-F) ump4Tg* to label keratinocytes [44], and *Tg(kdr:eGFP) s843Tg* to visualize blood vessels [45]. Adult zeb-rafish were housed in 3.5 L polycarbonate tanks connected to a recirculating system (Tecni-plast), at a density of 5–6 fish per liter and in the following conditions: salinity of 0.4%, conductivity of 400 μS, temperature of 28 ± 1˚C, and 12h/12h light/dark cycle. Adult fish were fed twice a day with Gemma Micro (Skretting, USA). Embryos were obtained from pairs of adult fish by natural spawning. They were collected in Petri dishes containing zebrafish water at 28˚C. From 5 dpf to juvenile stage, they were raised in water tanks and larvae were fed thrice a day with Gemma Micro and once a day with *Artemia salina* nauplii (NovoTom, JBL GmbH, Germany; from 10 dpf stage). Embryos and larvae were staged as described previously [46].

### RNA isolation and RT-PCRs

To determine the expression pattern of CEACAMz1 mRNA during development, 25–50 wild-type AB embryos/larvae ranging from 1-cell stage to 30 dpf stage were pooled and their total RNA was isolated with the NucleoSpin RNA kit (Macherey-Nagel). For tissue-specific distri-bution of CEACAMz1 mRNA, selected organs of 2–3 male or female adult zebrafish (AB strain) were dissected, pooled and their total RNA was extracted using TRIzol reagent (Gibco) [47]. cDNAs were then synthesized by reverse transcription of 500 ng of total RNA per reac-tion using the High Capacity cDNA Reverse Transcription kit (Applied Biosystems). RT-PCRs were performed with a cDNA amount equivalent to 5 ng total RNA using the following set of CEACAMz1-specific primers: 5'-CGTCTGAGGTCTGAGGAAGAAG-3' (forward) and 5'-

GGTGAAGTACACGGTGTCGTTC-3' (reverse). The expected size for the amplified fragment is 474 bp. The mRNA of the constitutively expressed Receptor for Activated C Kinase (RACK1) was used as a positive control in RT-PCR with primers: 5'-CCTCGCCAAAATGACC GAGC-3' (forward) and 5'-GGTGTACTTGCAGACTCCCAG-3' (reverse) (expected size of the amplified fragment: 433 bp).

## Cloning of *ceacamz1* open-reading frame and sequence analysis

The full-length DNA sequence coding for CEACAMz1 was amplified by PCR from 7 dpf zebrafish cDNA using the forward primer CEAz1fl_Fw1 (5'-CGTCTGAGGTCTGAGGAAGAAG-3') and the reverse primer CEAz1fl_Rv1 (5'-GCCTGAGAGCAGTCTCAGAATTAC-3') (see S1 Fig for their position on CEACAMz1 cDNA). The 3.1 kb PCR product was then inserted into pCR4Blunt-TOPO vector using the Zero Blunt TOPO PCR cloning kit for sequencing (Invitrogen) and fully sequenced (Eurofins, see S1 Fig for sequencing primers). The sequence of the corresponding protein, CEACAMz1, was then analyzed for secondary structure, sorting signals and post-translational modifications with the following prediction servers and software: SignalP 5.0 [48] for peptide signal prediction, PredGPI [49] for GPI-anchorage, TMHMM 2.0 [50] for membrane topology, NetNGlyc 1.0 for identification of glycosylation sites (http://www.cbs.dtu.dk/services/NetNGlyc/), HMMER 3.2.1 [51] to predict secondary structure motifs. Sequence alignments were performed with Clustal Omega [52] and analysis of the sequence conservation was done with ALINE [53].

## Whole-mount *in situ* hybridization (ISH) on zebrafish embryos and larvae

A fragment of *ceacamz1* cDNA ranging from the beginning of Exon 6 to the beginning of Exon 13 (S1 Fig) was amplified by PCR using primers 5'-TAGTGTGACCTTCAGCTGTT-3' and 5'-TGGAATCTGTCAGCTGCGTT-3' and cloned into pCR2.1 TOPO vector (Invitrogen) in either forward or reverse direction. The resulting plasmids were linearized with *Kpn* I and digoxigenin (DIG)-labeled (Roche, France) sense and antisense RNA probes were obtained by *in vitro* transcription using T7 RNA polymerase (Biolabs, France). Whole-mount ISH were performed according to the Thisse & Thisse protocol [54]. Briefly, embryos and larvae at the desired developmental stage were fixed in 4% paraformaldehyde, dehydrated in ethanol, digested with proteinase K (New England Biolabs) after rehydration, and fixed again. Pre-hybridization was performed at 65 ˚C in HM+ buffer (50% formamide, 5X saline sodium citrate buffer, 10 mM citric acid pH 6, 0.1% Tween 20, 0.5 mg/ml yeast tRNA, 0.05 mg/ml heparin) and followed by hybridization with 0.5 mg/ml DIG-labeled RNA probe (sense or antisense) in HM+ buffer. The probes were detected with alkaline phosphatase-conjugated anti-DIG antibodies (dilution 1/5000; Roche Applied Science) using nitro blue tetrazolium chloride/5-bromo-4-chloro-3-indolyl-phosphate, toluidine-salt (Roche Applied Science). After staining, embryos were mounted in 90% glycerol prior to imaging with an Olympus MVX10 epifluorescence stereomicroscope equipped with a digital color camera (Olympus XC50). For each stage and probe, ISH were performed on at least 5 distinct embryos/larvae.

## Generation of a CEACAMz1 fluorescent reporter line

The *Tg(ceacamz1:mCherry-F) ump9Tg* transgenic reporter line was generated following previously reported procedures [44]. A 7 kb fragment of the *ceacamz1* promoter was amplified from zebrafish genomic DNA using forward primer CEAz1P0 (5'-GAACCACTTAAGGCTA CCACA-3') and reverse primer CEAz1E1N2 (5'- ATATGCGGCCGCCAAGAACTTTAAATCC CATTTTGGAA-3'), which extends 20 bp downstream *ceacamz1* AUG. The 7284 bp PCR fragment was then phosphorylated, digested with *Not* I and ligated in pTol2BNmCherry-F vector,

just upstream of the reading frame of farnesylated mCherry (mCherry-F) protein so that the *ceacamz1* start codon would be in frame with the downstream mCherry-F open reading frame. The plasmid was injected into one-cell stage embryos (wild-type AB zebrafish line) together with the Tol2 transposase RNA [55]. F0-microinjected embryos/larvae were screened for transgene expression at 2 and 5 dpf, and stable lines were established.

## Immunocytochemistry

Immunocytochemistry was performed on 4 or 22 dpf larvae from the *Tg(ceacamz1:mCherry-F)* transgenic reporter line. For co-staining with concanavalin A (conA), live larvae were incubated for 2 hours in fish water supplemented with 50 μg.ml$^{-1}$ of Alexa Fluor 488-conjugated conA (Fisher Scientific) (4 larvae per well containing 500 μl fish water ± conA, in 24-well plates). Larvae were then washed thrice in fish water to remove unbound conA (10 min incubation for each wash), and they were anesthetized with 0.016% (w/v) tricaine before being immobilized in low melting-point agarose for imaging by confocal microscopy. For co-labelling with anti-Na$^+$/K$^+$-ATPase or anti-vH$^+$-ATPase antibodies, at least 10 larvae per condition were first fixed with 4% paraformaldehyde in PBS at 4˚C overnight. Fixed larvae were then washed thrice in PBS with 0.1% Tween 20 (PBT) and incubated for 10 min at room temperature with 0.3% (v/v) Triton X100 in PBT for permeabilization. After washing again in PBT, the larvae were incubated for 5 hours at room temperature in blocking solution (1% BSA, 1% DMSO, 2% lamb serum in PBS) and then for 24 to 48 hours at 4˚C with an antibody against the α subunit of eel (*Anguilla japonica*) Na$^+$/K$^+$-ATPase (polyclonal, produced in rabbit [56]) diluted 1:1000 in blocking solution or with an antibody against the B subunit of dace (*Tribolodon hakonensis*) vacuolar-type H$^+$-ATPase (polyclonal, produced in rabbit [57]) diluted 1:500 in blocking solution. These antibodies were a generous gift from Prof. Shigehisa Hirose and Dr. Nobuhiro Nakamura (Tokyo Institute of Technology, Yokohama, Japan). Following extensive washes in PBT, the larvae were then incubated overnight at 4˚C with goat anti-rabbit IgG antibody conjugated with Alexa Fluor 488 (Invitrogen, diluted 1:500 in blocking solution). After washing again extensively in PBT, larvae were immobilized in low melting-point agarose for imaging by confocal microscopy.

## Confocal microscopy and image analysis

Prior to live imaging using confocal microscopy, larvae were anesthetized with 0.016% (w/v) tricaine, mounted in 35 mm glass-bottom dishes (FluoroDish, World Precision Instruments, UK) and immobilized with 1% low-melting point agarose. Fluorescence stacks and bright-field images were acquired at 28˚C using a spinning disk Nikon Ti Andor CSU-W1 microscope (10x/0.25 Air, 20x/0.75 Air or 40x/1.15 Water objectives). Images were processed with Fiji (Image J software) and compressed into maximum intensity Z-projections. 3D-reconstructions were generated with IMARIS software (https://imaris.oxinst.com/). Image acquisition and processing were performed on the Montpellier RIO Imaging microscopy platform (MRI).

# Results

## Sequence analysis of CEACAMz1

To date, up to ten distinct CEACAM putative genes have been identified in zebrafish based on genomic sequence comparisons [4,5]. Nine of these genes appear to be clustered on chromosome 16 whereas the remaining gene is located on chromosome 19. Most of these genes have however not been further validated and data on gene expression are still lacking. Only one of these genes is currently annotated as a CEACAM orthologue in the ZFIN database and is

referred to as *ceacam1* or *ceacam5*, based on its closest sequence similarity with these two human CEACAMs. Nevertheless, sequence identity with any of the human genes remains quite low (see below). We therefore decided to adopt a distinct nomenclature for zebrafish genes and we renamed this gene *ceacamz1* to avoid confusion with the human genes. Interestingly, synteny mapping revealed that on zebrafish chromosome 16, the *ceacamz1* gene is surrounded by orthologs of *atp1a3* (Na$^+$/K+-ATPase subunit alpha3), *grik5* (glutamate ionotropic receptor kainate type subunit 5), *rabac1* (prenylated Rab acceptor 1), *gsk3a* (glycogen synthase kinase 3 alpha), *dedd1/2* (death effector domain-containing 1/2), *znf574* (zinc finger protein 574), *cnfn* (cornifelin), *ethe1* (persulfide dioxygenase) and *pafah1b3* (platelet activating factor acetylhydrolase 1b catalytic subunit 3) [4,58]. These genes are all found in the q13.2 region of human chromosome 19 which encodes all twelve human *ceacam* genes. *Ceacamz1* thus shares the same genomic context as its human counterparts.

The *ceacamz1* gene is 35 kb long and contains 13 exons (Fig 2A). To further characterize this gene and its product, CEACAMz1, we first cloned its full-length coding DNA sequence from a cDNA library prepared from total RNAs of 7 days post-fertilization (dpf) zebrafish larvae. The complete *ceacamz1* cDNA sequence is shown in S1 Fig and the corresponding protein sequence is displayed in Fig 2B. We observe a few variations in our CEACAMz1 protein as compared to the sequence reported in the database but most of these lead to conservative or semi-conservative amino acid replacements and reflect the polymorphism of the laboratory zebrafish strains. Similarly to the human CEACAMs, CEACAMz1 is predicted to be exclusively composed of Ig motifs with an N-terminal Ig-V domain followed by ten Ig-C2 domains alternating between A and B types (Fig 1). Large genes containing several Ig-modules have been generated by gene duplication/transfer throughout evolution, according to the exon shuffling theory proposed by Gilbert [59]. These processes have been facilitated within the Ig superfamily due to the fact that Ig domains are generally encoded by a single, symmetrical exon, i.e. introns placed at the boundaries of each Ig-coding exon are in the same phase [60]. Such property indeed allows for efficient exon shuffling without introducing a frame shift in the downstream sequence of the modified gene, in accordance with the phase-compatibility rule [61,62]. In agreement with this principle, each Ig domain of the CEACAMz1 protein is encoded by a single exon and all introns appear to be phase 1 introns (S1 Fig).

Analyses with various prediction servers indicate that CEACAMz1 is a secreted protein anchored in the plasma membrane via GPI. The CEACAMz1 secretory signal is most probably cleaved after residue Cys18, with an alternative cleavage site proposed between residues Val20 and Asn21 (Fig 2B). The ω-site for GPI-anchoring is unambiguously identified as Asn999. The mature CEACAMz1 protein is therefore predicted to be 981 residues long, which represents a molecular weight of 106 kDa in its bare form. The native protein is however expected to be much larger since its ectodomain should be heavily glycosylated, as observed for human CEACAMs, with up to 38 potential glycosylation sites identified. CEACAMz1 is thus much larger than any of the human CEACAM proteins.

Traditionally, the N-terminal domain of receptors belonging to the Ig superfamily is thought to bear the main binding site(s) for specific ligands. Although this concept starts being questioned in light of recent studies demonstrating the role of constant domains in ligand recognition for Ig-containing receptors, including CEACAMs [63], the comparison of CEACAMz1 N-terminal domain with that of human CEACAMs may still reveal informative to predict functions for the zebrafish receptor. As shown in Fig 2C, sequence alignment of CEACAMz1 N-terminal domain with the corresponding domain of all twelve human CEACAMs shows a weak degree of conservation, with a maximal sequence identity of 27% reached with human CEACAM3. CEACAMz1 N-terminal domain also contains important insertions and deletions as compared to its human counterparts. In fact, purely based on sequence analysis,

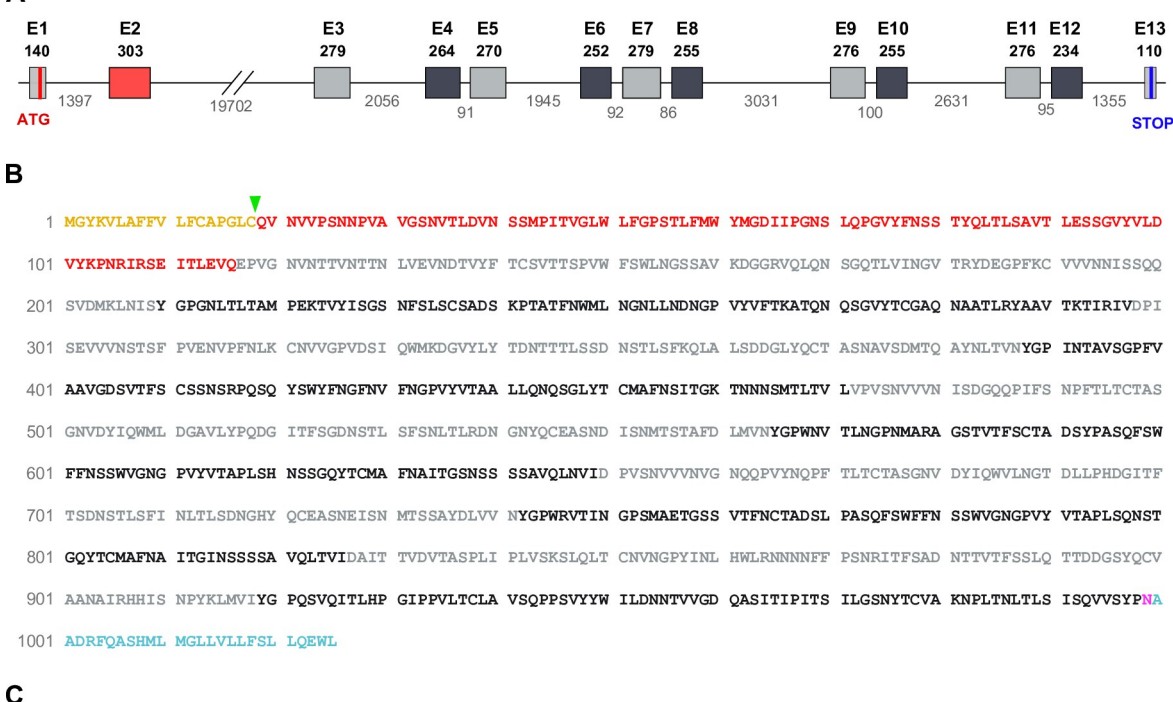

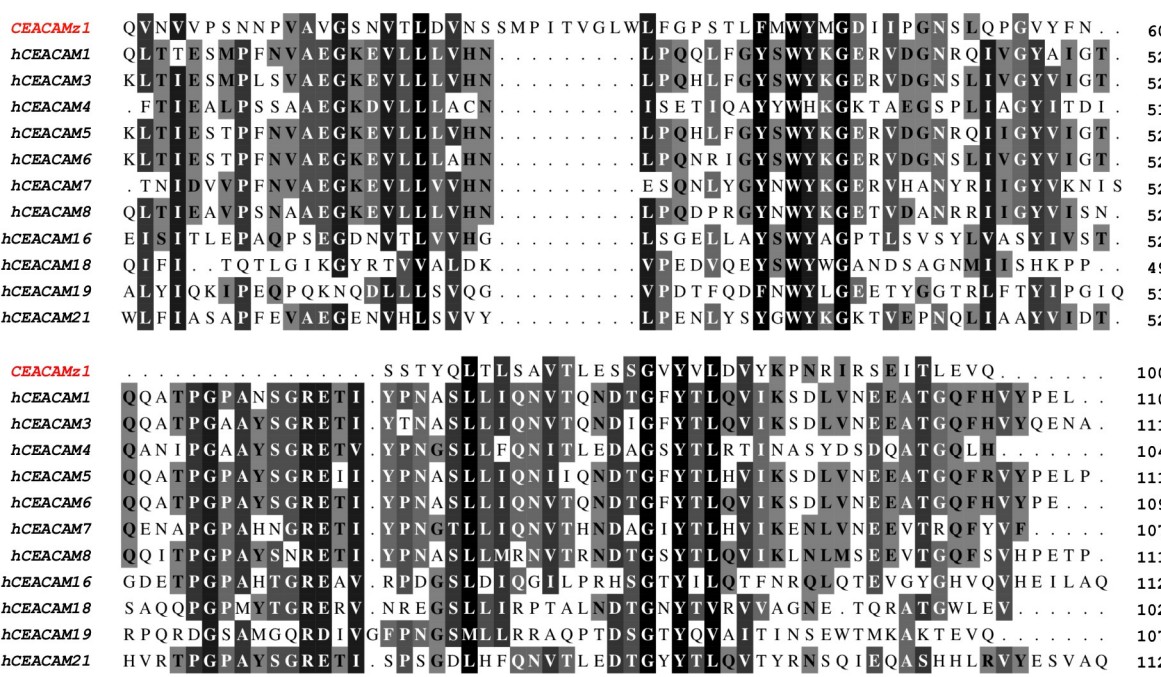

**Fig 2. Sequence analysis of the *ceacamz1* gene and its product. A.** Organization of *ceacamz1* coding region. The 35 kb-long *ceacamz1* transcription unit is composed of 13 exons (red and grey boxes). The length in base pairs of the different exons and introns is indicated on the scheme. Each single Ig domain is encoded by a single exon (color-coded as in Fig 1). **B.** Protein sequence of CEACAMz1. The putative signal peptide is highlighted in orange and the green arrowhead indicates the most probable cleavage site (after Cys18). The Ig-V N-terminal domain is indicated in red. The Ig-C2 domains are highlighted in light grey (A-type) or dark grey (B-type). The ω-site for GPI-anchoring (Asn999) is displayed in purple. The residues highlighted in blue are cleaved off in the mature protein. **C.** Sequence alignment of the N-terminal Ig-V domain of CEACAMz1 with that of the twelve human CEACAMs. Amino acids are shaded from black to light grey according to their degree of conservation.

CEACAMz1 N-terminal domain does not appear to be a canonical variable domain. It may rather belong to the I-type of Ig domains (intermediate fold between variable and constant domains), an Ig subset encountered in several cell adhesion molecules [64]. Without structural information on the protein, the classification in either V- or I-type remains however uncertain for this domain. In any case, no strict orthology can be established between CEACAMz1 and any of the human CEACAMs.

## Temporal and tissue-specific distribution of CEACAMz1 mRNA in zebrafish

As we could not get hint on CEACAMz1 function based on its sequence comparison with the human receptors, we first analyzed the distribution of its mRNA in larvae and adult zebrafish. Using RT-PCR, we evaluated the presence of CEACAMz1 mRNA in various tissues of adult zebrafish. Expression was predominantly detected in gills whereas low expression levels were also measured in several tissues including genital organs, brain, heart, muscle, and possibly kidney and intestine (Fig 3A). In comparison, mRNA from the ubiquitous RACK1 protein was expressed in high amounts in all tissues studied. Next, we determined CEACAMz1 mRNA expression during embryogenesis. A CEACAMz1-specific RT-PCR product could be detected starting from 18 hours post fertilization (hpf) and thereafter (Fig 3B). No CEA-CAMz1-specific fragment was amplified at 14 hpf or before, suggesting that *ceacamz1* expression is not turned on before the end of the first day.

## Expression profile of CEACAMz1 in zebrafish larvae

Gills are formed quite tardily in zebrafish. They do not become fully functional for oxygen uptake and ionic exchanges until at least 14 dpf, their full maturity being reached after 21 dpf [65]. Since its mRNA already appears at the end of the first day, we wanted to determine in which tissue(s) CEACAMz1 expression occurs at early larval stages, before gills development is complete. For this purpose, we performed whole-mount *in situ* hybridization (ISH) on zebrafish embryos and larvae ranging from 1 to 6 dpf (Fig 4). In agreement with the RT-PCR experiments, a specific signal for CEACAMz1 mRNA is readily detected at 1 dpf stage (Fig 4A and 4B). CEACAMz1-positive cells accumulate on the surface of the embryos, mainly around the yolk sac and on the most rostral part of the yolk extension (Fig 4B). In contrast, no signal is detected using a sense RNA probe (Fig 4C). At later stages (Fig 4D–4K), CEACAMz1-positive cells are still predominantly visible on the ventral side of the larvae and on the yolk extension, with labelled cells extending further towards the tail. From 3dpf and later on, CEACAMz1-positive signal also appears increasingly around the forming gills (Fig 4G–4K). In all these experiments, CEACAMz1-expressing cells seem to be localized in the upper epidermal layers of zebrafish skin.

To further investigate the localization and nature of CEACAMz1-expressing cells, we generated a transgenic reporter line expressing the red fluorescent protein mCherry under the control of the *ceacamz1* promoter. For this purpose, a 7 kb DNA fragment, upstream of the CEACAMz1 coding region, was amplified by PCR from zebrafish genomic DNA and was used to direct mCherry expression in zebrafish. Similarly to what we observe in our ISH experiments, CEACAMz1-expressing cells localize on the ventral region of the zebrafish epidermis (Fig 5A). At 4 dpf, these cells are mostly present on the yolk sac and yolk extension, they are also present to a lesser extent around the gill region (Fig 5A). Within our transgenic line, the expression of the mCherry protein is driven to the membrane thanks to the addition of a farnesylation motif at the carboxyl terminus of the protein. This allows us to directly visualize the contour, and therefore the morphology, of the fluorescent cells. CEACAMz1-positive cells

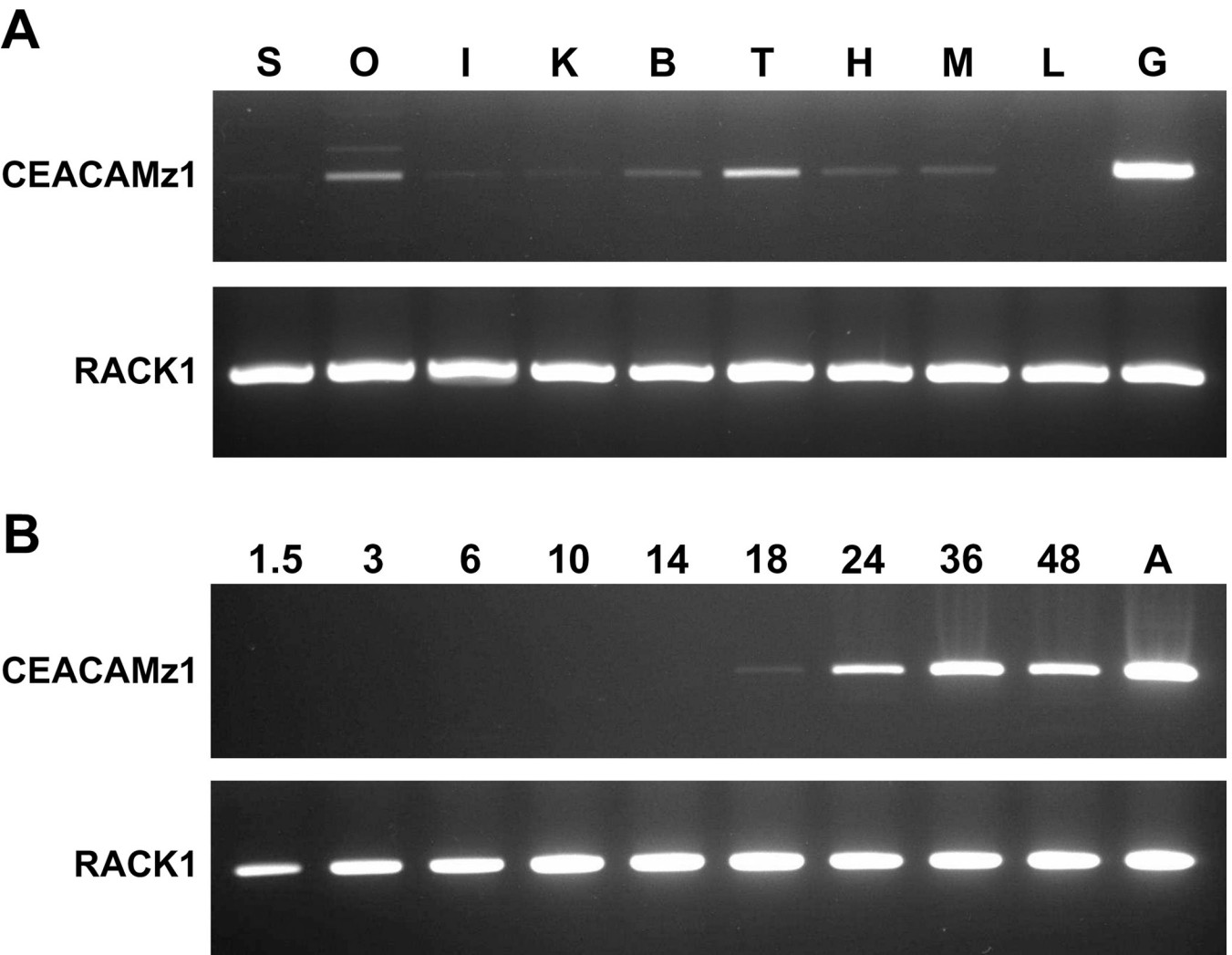

**Fig 3. Spatial and temporal distribution of *ceacamz1* mRNA. A.** RT-PCR analysis of *ceacamz1* expression in various tissues of adult zebrafish. S: Skin; O: Ovary; I: Intestine; K: Kidney; B: Brain; T: Testis; H: Heart; M: Muscle; L: Liver; G: Gills. **B.** RT-PCR analysis of *ceacamz1* expression during development. The developmental stages analyzed are indicated in hpf above each lane. "A" refers to adult stage (> 30 dpf). The constitutive *rack1* gene was used as control in both sets of experiments.

have an irregular shape, some being quite roundish while others display sharper edges (Fig 5B). Several cells also possess thin extensions resembling dendrites. They do not form a contiguous patch on the fish surface. Instead, they are well separated from each other and no cell-cell contacts can be observed within this population (Fig 5B).

RT-PCRs revealed that at the mRNA level, *ceacamz1* expression is turned on sometime between 14 and 18 hpf. To evaluate at which stage *ceacamz1* expression can be detected at the protein level, we monitored the apparition of the mCherry signal in our *Tg(ceacamz1: mCherry-F)* reporter line by imaging zebrafish larvae every hour starting from 14 hpf stage. As depicted in S2 Fig, mCherry signal can be readily detected at 17 hpf (16 somites stage). Knowing that there might be one or two hours delay between mRNA apparition and actual protein translation, and taking into account that early fluorescence signal may not be intense enough to be detected by confocal microscopy, these observations allow us to narrow down the apparition of *ceacamz1* mRNA to a time frame most probably comprised between 14 and 16 hpf.

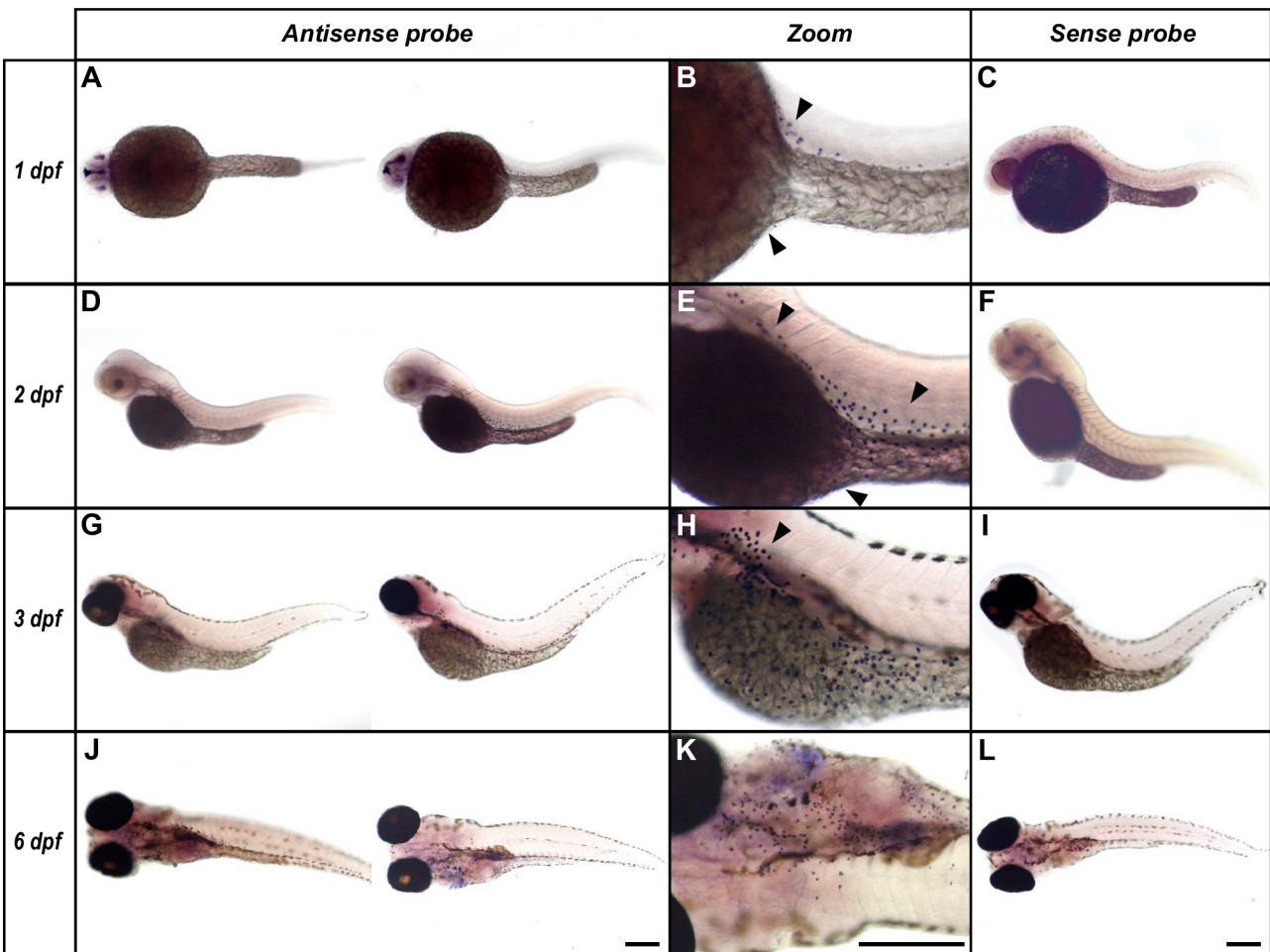

**Fig 4. Whole mount *in situ* hybridization with specific zebrafish *ceacamz1* RNA probes.** Larvae at 1 dpf (**A-C**), 2 dpf (**D-F**), 3 dpf (**G-I**) or 6 dpf (**J-L**). *Ceacamz1* mRNA signals are detected in discrete cells around the yolk sac and yolk extension readily from 1 dpf stage (as indicated by black arrows). They also appear around the forming gills after 3 dpf stage. Scale bars: 1 mm.

This also suggests that very rapidly, if not right from the beginning of its expression, CEA-CAMz1 localizes in the same type of cells from the ventral epidermis as those identified later on during development (Fig 5A).

At larval stage, the zebrafish epidermis adopts a bilayered structure [66]. The most superficial layer is composed of a continuous pavement of keratinocytes, between which are intercalated mucous cells and ionocytes (Fig 5C). The stratum just below is formed by undifferentiated basal cells. To evaluate in which epidermal layer the CEACAMz1-expressing cells are anchored, we crossed our *Tg(ceacamz1:mCherry-F)* line with the *Tg(il1b:eGFP-F)* line that expresses green fluorescent protein under the control of the interleukin 1B-specific promoter [44]. It was previously shown that keratinocytes express high levels of IL1B at early larval stages [44]. Imaging of 4dpf *Tg(ceacamz1:mCherry-F)*x*Tg(il1b:eGFP-F)* larvae indeed allowed us to visualize the pavement-like network of keratinocytes on the zebrafish skin (Fig 5D). We also observed that CEACAMz1-expressing cells are generally present at the junction between two or three contiguous keratinocytes (Fig 5D). To assess whether CEACAMz1-positive cells are contained in the same layer as keratinocytes or whether they are deeper buried within zebrafish epidermis, we generated a 3D-reconstruction with the IMARIS software. As

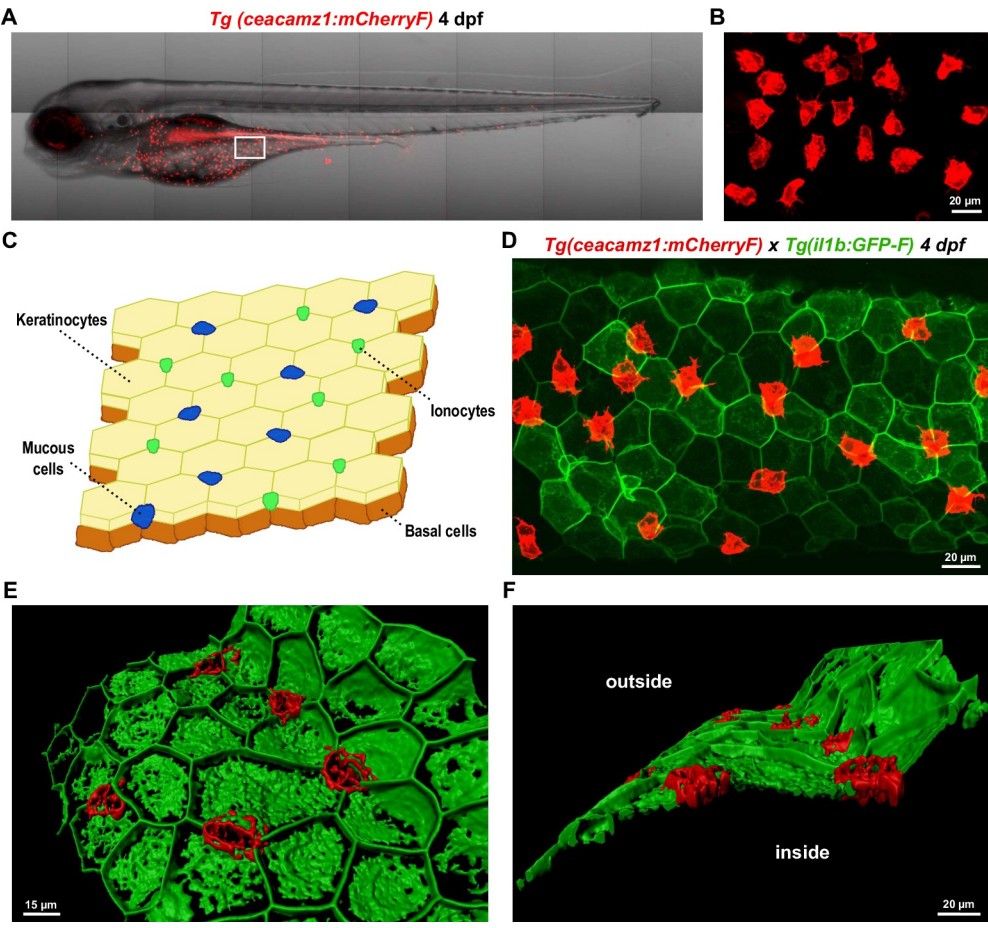

**Fig 5. Expression profile of CEACAMz1 on the zebrafish epidermis using the fluorescent reporter line *Tg (ceacamz1:mCherry-F)*. A.** Live imaging of a 4 dpf *Tg(ceacamz1:mCherry-F)* zebrafish larva using spinning disk confocal microscopy. **B.** Zoom on the ventral region of the 4 dpf transgenic larva as indicated in panel A. **C.** Schematic representation of the cell composition of zebrafish skin epidermis at early larval stages. **D.** Z-projection of a portion of the ventral skin of a 4 dpf double-crossed *Tg(ceacamz1:mCherry-F)* x *Tg(il1b:eGFP-F)* zebrafish larva. CEACAMz1-expressing cells intercalate within the pavement of keratinocytes. **E.** and **F.** 3D-reconstructions of the Z-projection shown in panel D using IMARIS. Top view (**E**) and lateral view (**F**). CEACAMz1-expressing cells are embedded within the same layer as keratinocytes and they protrude on both sides of this layer.

seen in Fig 5E and 5F, mCherry-labelled cells are embedded directly within the layer of keratinocytes and they protrude on both sides of this layer. In conclusion, CEACAMz1-expressing cells are present in the uppermost stratum of the larval zebrafish epidermis, between keratinocytes from the ventral region, and they are in direct contact with the external environment.

## Identification of the cell population expressing CEACAMz1

Two different types of cells intercalate between keratinocytes on zebrafish skin: mucous cells and ionocytes (Fig 5C). Mucous cells have a quite widespread distribution all over the fish skin [67]. The restricted expression profile of CEACAMz1-positive cells suggests that they rather belong to the ionocyte category. Five different subtypes of ionocytes have been characterized in zebrafish [68]. They all display specific expression patterns. At larval stages, K$^+$-secreting ionocytes (KS cells), ionocytes expressing the Na$^+$/Cl$^-$ co-transporter (NCC cells) and ionocytes rich in Na$^+$/K$^+$-ATPase (NaR cells) have a relatively uniform distribution over the zebrafish

epidermis [69–71]. In contrast, ionocytes expressing the solute carrier 26 (SLC26 cells) localize mainly in the heart, mesonephros, neuromast and gills [72], while ionocytes rich in $H^+$-ATPase (HR cells) restrict to the ventral side of the larvae [73]. Both NCC, NaR and HR cells are present around the yolk sac and the forming gills but NCC ionocytes seem to appear more tardily than the others in these regions [70]. CEACAMz1-positive cells are readily visible on the yolk sac before the end of the first day, suggesting that they belong to either the NaR or the HR subtype of ionocytes. To verify this hypothesis, we performed immunolabelling on 4 dpf *Tg(ceacamz1: mCherry-F)* larvae using either an antibody against the α-subunit of eel $Na^+/K^+$-ATPase (labelling of NaR cells) or an antibody against the B subunit of dace, vacuolar-type $H^+$-ATPase (labelling of HR cells) [73]. Cells labelled by the anti-$Na^+/K^+$-ATPase antibody are clearly distinct from the CEACAMz1-positive cells (Fig 6A–6C). In contrast, the epidermal cells expressing CEACAMz1 are all labelled by the anti-v$H^+$-ATPase antibody (Fig 6D–6F). To confirm these findings, we also performed co-immunostaining with concanavalin A (conA). This lectin was shown to preferentially label the apical membrane of HR cells [73,74]. We indeed observed a perfect co-localization of the red fluorescent signal arising from our mCherry-labelled CEACAMz1 cells with the green fluorescent signal of Alexa Fluor 488-conjugated conA (Fig 6G–6I). All CEACAMz1-positive cells were labelled with both anti-v$H^+$-ATPase antibody and conA. None of them displayed co-labelling with anti-$Na^+/K^+$-ATPase antibody. Taken together, these data suggest that CEACAMz1 is expressed exclusively in HR ionocytes.

## Localization of CEACAMz1-expressing ionocytes in the gills at late larval stages

Our RT-PCR experiments revealed that CEACAMz1 is no longer detected in the skin by adult stage. Instead, it is almost exclusively present in the gills (Fig 3A). To determine when the transition from skin to branchial expression occurs, we imaged our *Tg(ceacamz1:mCherry-F)* line at different developmental stages ranging from 4 to 32 dpf (Fig 7). As shown previously, CEACAMz1 expression restricts mainly to the ventral portion of the zebrafish skin at 4 dpf, with almost no signal detected in the gill region (Fig 7A and 7B). At 8 dpf, CEACAMz1 expression is still predominant on the ventral epidermis (Fig 7D), but labelling around the branchial arches is also clearly visible (Fig 7C). At later stages, CEACAMz1-positive cells become more and more abundant in the gills (Fig 7E, 7G and 7I). From 21 dpf and later on, they accumulate mainly along the gill filaments and possibly the branchial arches (Fig 7G–7I). Expression of CEACAMz1 on the ventral epidermis remains important at 15 dpf (Fig 7F). In contrast, the number of CEACAMz1-positive cells decreases significantly on the skin at 21 dpf (Fig 7H) and these cells have almost totally disappeared from the ventral region at 32 dpf (Fig 7J). This suggests that the transition from cutaneous to branchial expression of CEACAMz1 occurs sometime between 21 and 32 dpf. Co-immunostaining on 22 dpf *Tg(ceacamz1:mCherryF)* with conA revealed again a perfect overlap between CEACAMz1-expressing cells and the cells labelled by the lectin (Fig 6J–6L). These results demonstrate that later stage expression of CEACAMz1 in the gills still occurs predominantly in HR-ionocytes.

At late larval stages, CEACAMz1-positive ionocytes have a radial distribution which seems to follow the gill filaments (Fig 7G and 7I). To characterize the localization of these cells within gills in more details, we crossed our *Tg(ceacamz1:mCherry-F)* line with the *Tg(kdr:eGFP)* line which drives the expression of GFP in the zebrafish vasculature [45]. We then imaged double-transgenic larvae at 22 dpf stage, focusing on the most posterior part of the gills (Fig 8A). As seen in Fig 8B, GFP fluorescence allows to clearly visualize blood vessels in the branchial arch as well as filament and lamellar arteries. CEACAMz1-ionocytes are distributed mostly along the filament arteries and to a minor extent, along the branchial vessels. To confirm these

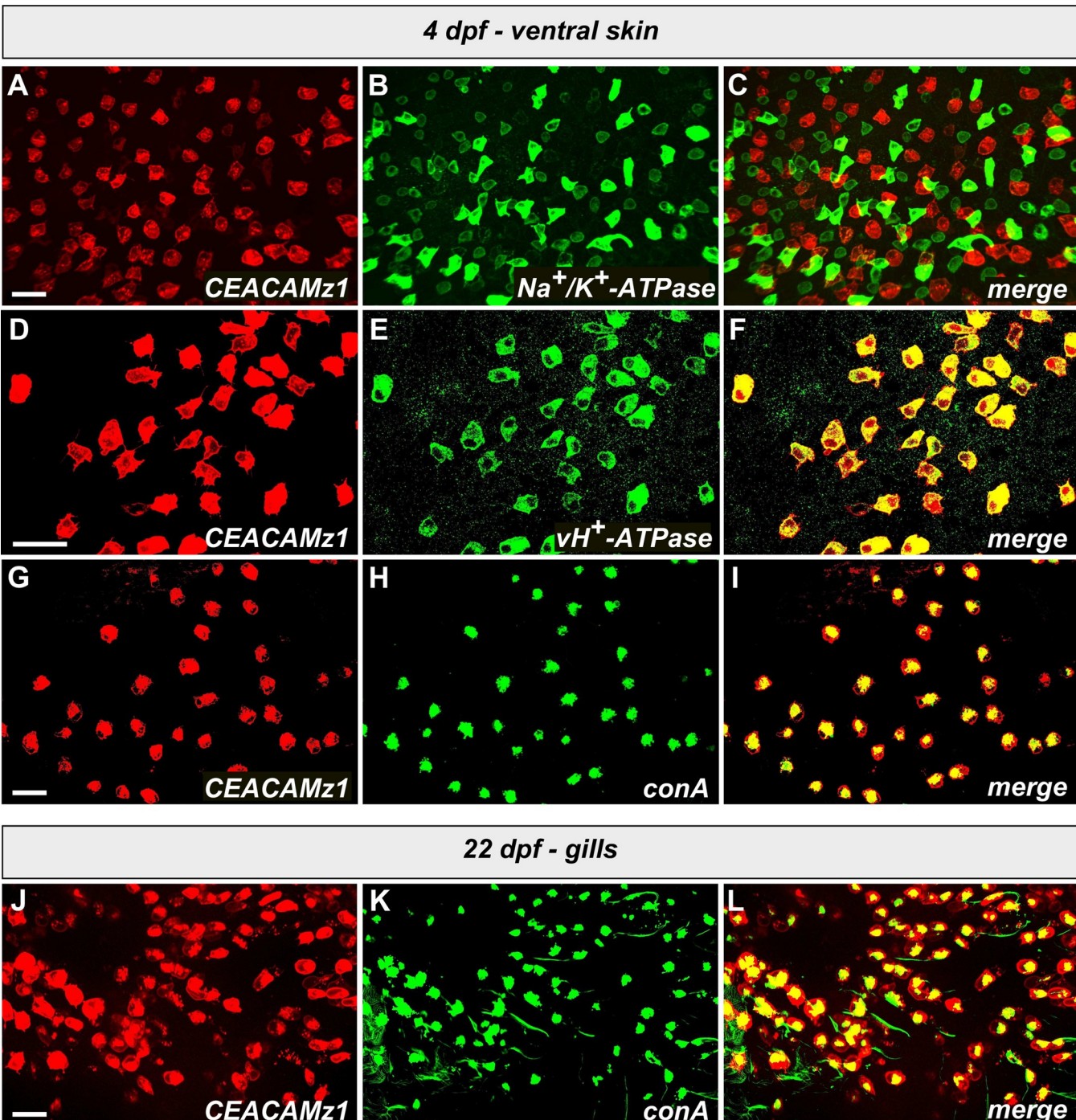

**Fig 6. Identification of the ionocyte subtype that expresses CEACAMz1 on zebrafish epidermis. A-C.** High magnification view of the yolk sac surface of a 4 dpf *Tg(ceacamz1:mCherry-F)* larva (**A**) immunolabelled with anti-Na$^+$/K$^+$-ATPase antibody (**B**); merge of A and B (**C**). **D-F.** High magnification view of the yolk sac surface of a 4 dpf *Tg(ceacamz1:mCherry-F)* larva (**D**) immunolabelled with anti-vH$^+$-ATPase antibody (**E**); merge of D and E (**F**). **G-I.** High magnification view of the yolk sac surface of a 4 dpf *Tg(ceacamz1:mCherry-F)* larva (**G**) co-stained with Alexa Fluor 488-conjugated conA (**H**); merge of G and H (**I**). (**J-L**) High magnification view of the gill region of a 22 dpf *Tg(ceacamz1:mCherry-F)* larva (**J**) co-stained with Alexa Fluor 488-conjugated conA (**K**); merge of J and K (**L**). At all stages analyzed, CEACAMz1-expressing cells perfectly co-localize with cells labelled by the vH$^+$-ATPase antibody or by conA. They do not overlap at all with the cells labelled by the Na$^+$/K$^+$-ATPase antibody.

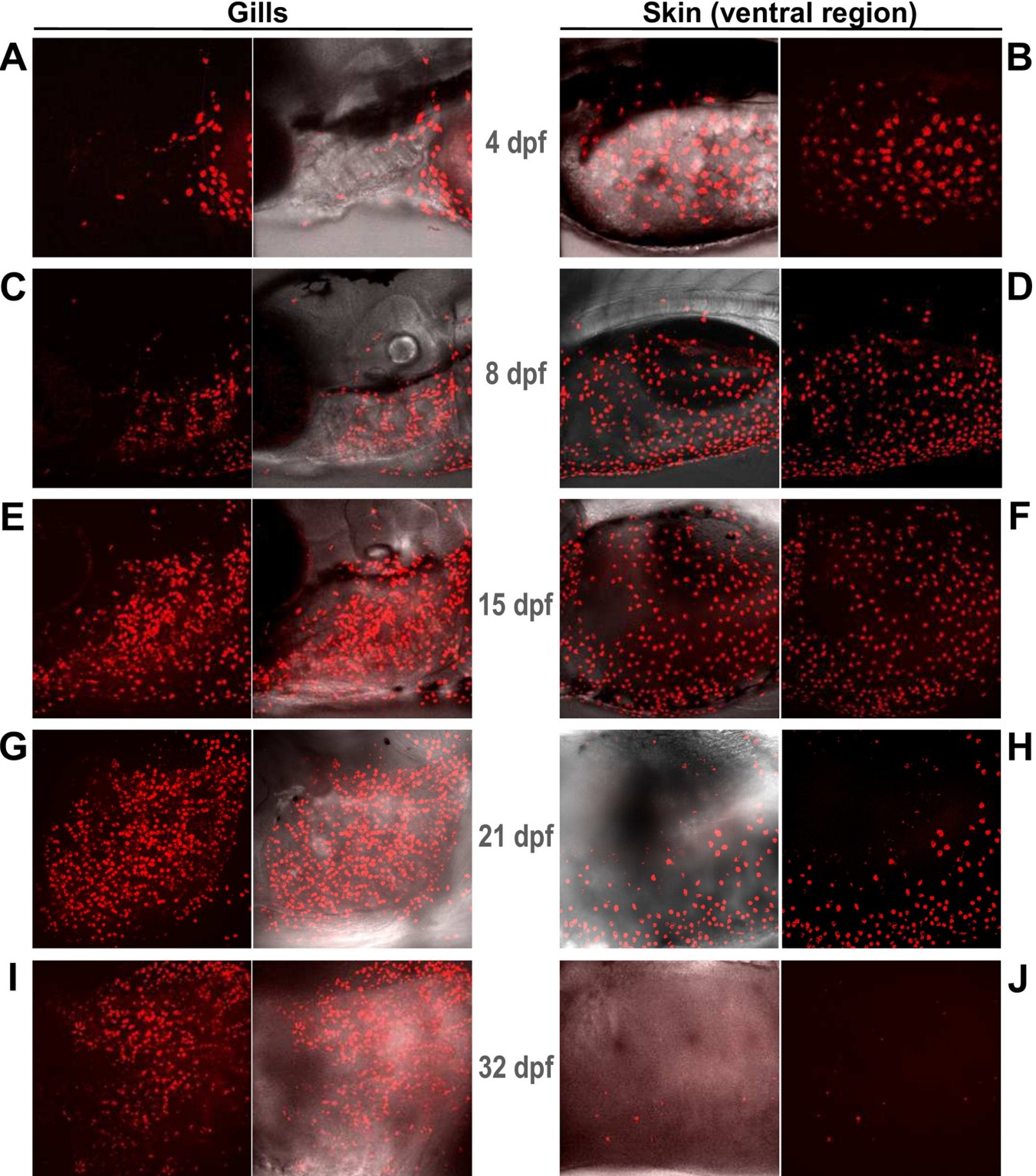

**Fig 7. Evolution of CEACAMz1 expression in gills and on ventral skin at late larval stages.** Live imaging of larvae from the *Tg(ceacamz1:mCherry-F)* reporter line using spinning disk confocal microscopy at various developmental stages: 4 dpf (**A-B**), 8 dpf (**C-D**), 15 dpf (**E-F**), 21 dpf (**G-H**), 32 dpf (**I-J**). Visualization around the gill region (**A,C,E,G,I**) or on the ventral skin region (**B,D,F,H,J**).

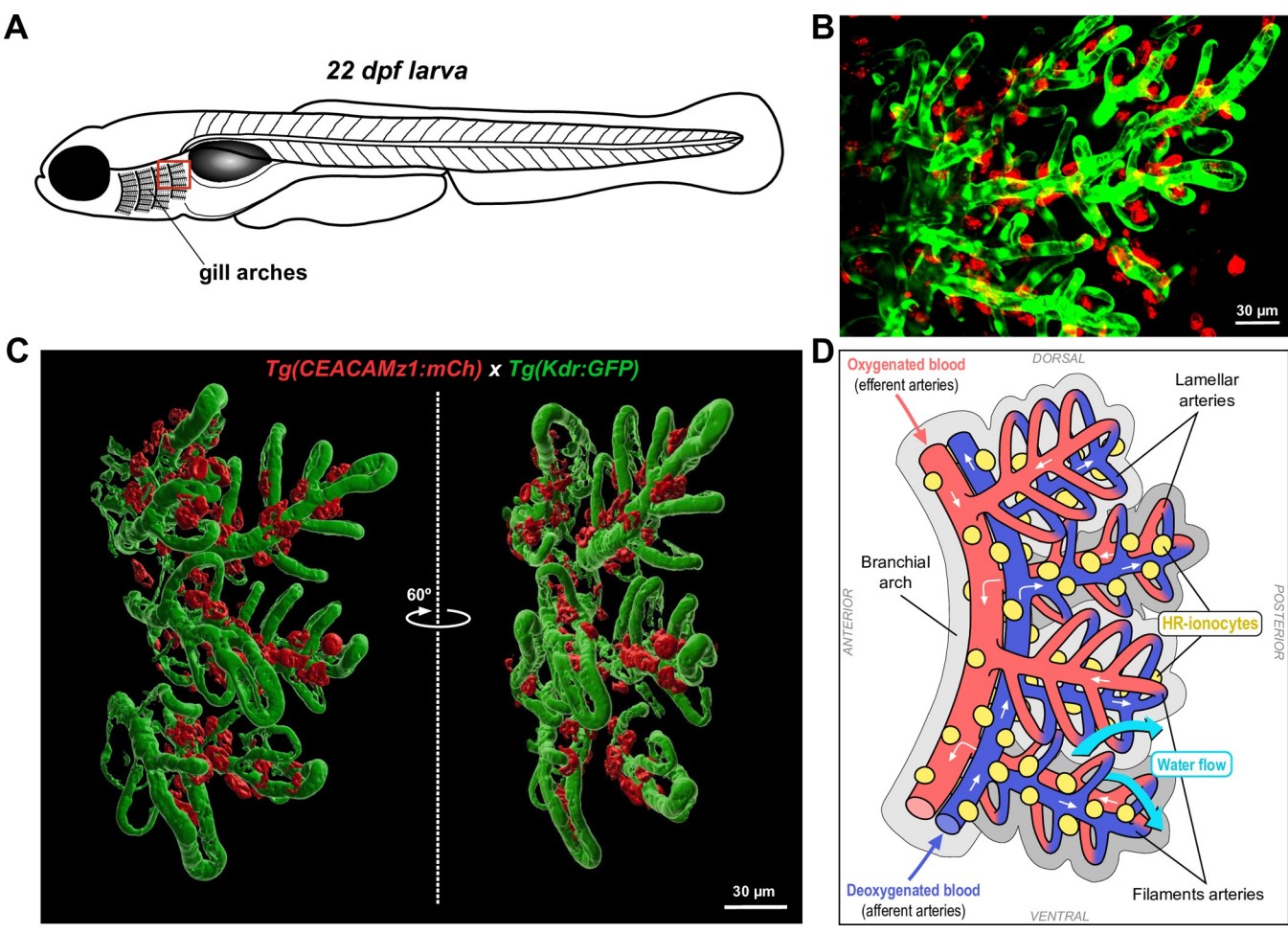

**Fig 8. Distribution of CEACAMz1-positive ionocytes with respect to vasculature in the gills at late larval stages. A.** Schematic representation of a juvenile zebrafish larva to highlight the part of the gills on which imaging was performed. **B.** Live imaging of a 22 dpf double-crossed *Tg(ceacamz1:mCherry-F)* x *Tg(kdr:eGFP-F)* larva using spinning-disk confocal microscopy. Z-projection centred on the most posterior arch of the gills. CEACAMz1-ionocytes (red) distribute along the branchial blood vessels (green), mainly the filament arteries at this stage. **C.** 3D reconstruction of the most posterior branchial arch using IMARIS software. The two distinct views are rotated by 60˚. CEACAMz1-expressing cells are localized along the filament arteries on the inner side of the branchial arch, i.e. along the afferent arteries. **D.** Schematic representation of the 3D reconstruction visualized in C. Blood vessels are color-coded according to their oxygen content (efferent arteries in red, afferent arteries in blue). HR ionocytes expressing CEACAMz1 are displayed in yellow. The direction of water flow is indicated with blue arrows and is opposite to blood flow (white arrows).

observations, we performed a 3D-reconstruction of the last branchial arch at 22 dpf stage (Fig 8C). This 3D model clearly shows that the cells expressing CEACAMz1 are mainly located alongside the filament arteries. They do not appear to distribute within the lamellae, at least not at this stage. Interestingly, the 3D model also reveals that CEACAMz1-ionocytes concentrate on the inner side of the gill filaments, along the afferent blood vessels (Fig 8C and 8D). In conclusion, CEACAMz1-positive cells start to appear within the gills between 4 and 8 dpf stage, first in the branchial arches and later on in the gill filaments, whereas they disappear from the ventral skin between 21 and 32 dpf stage, when juvenile stage is reached.

## Discussion

In this paper, we provide the first description of a putative member of the CEACAM receptor family in zebrafish, CEACAMz1. Our analysis of the *ceacamz1* gene product predicts a large,

GPI-anchored extracellular protein composed of eleven Ig domains. Sequence comparison with the human proteins does not reveal any direct orthology with a specific member of the CEACAM family in humans, not even for the N-terminal Ig-V domain which is highly conserved among the human receptors. To assess whether CEACAMz1 could still display functional orthology with the human CEACAMs, we analyzed its spatial and temporal distribution in zebrafish larvae and adults. Using a fluorescent zebrafish reporter line for *ceacamz1* expression, ISH experiments and RT-PCR analyses, we showed that CEACAMz1 is first expressed on the ventral skin of the fish at early larval stages, mainly on the yolk sac surface and along the yolk extension. CEACAMz1-positive cells readily appear on the skin at 17 hpf. After a few days of development, CEACAMz1-positive cells are also detected in the forming gills. Expression on the skin remains important until juvenile stage, at which point there is no more detectable *ceacamz1* expression in this tissue. After one month age, CEACAMz1 is almost exclusively present in the gills and this is still the case during adulthood. We further show that the cell population expressing CEACAMz1 corresponds to ionocytes intercalated between pavement cells, on the uppermost layer of the zebrafish epidermis. We unambiguously identified them as HR ionocytes, using immunolabelling with anti-vH$^+$-ATPase antibody and conA lectin. Finally, we observe that at late larval stages, when CEACAMz1 expression is predominantly branchial, CEACAMz1-positive ionocytes accumulate on the inner face of the gill filaments, along the afferent arteries.

In adult fish, gills perform a variety of functions to allow for survival in an aquatic environment, including gas and ionic exchanges, maintenance of the acid-base balance and excretion of nitrogen waste [75,76]. Gills form however quite tardily during zebrafish development [46,65]. Primary gill filaments only appear at 3 dpf and gills do not become functional for both oxygen uptake and ionoregulation until at least 14 dpf [46,65,77]. In absence of a fully mature respiratory organ, gas and ion exchanges take place on the larval skin [78,79]. Zebrafish larvae completely rely on cutaneous respiration until at least 7 dpf. The shift to fully branchial respiration then gradually occurs until around 21 dpf, at which stage the gill respiratory surface has increased to a maximal size [80,81]. Ion exchanges are orchestrated by different subtypes of mitochondria-rich cells (MRCs), or ionocytes, that express different sets of transmembrane transporters to insure ion homeostasis [68,82,83]. As a consequence, the localization of these ionocytes coincides with the active sites for ionoregulation and respiration throughout development, i.e. they are first present on the skin and later on in the gills [74,84,85]. Interestingly, although it has long been assumed that gills developed in fish due to the pressure for an oxygen uptake function, it was more recently shown that ionoregulation is the first function to take place in the developing gills [65,86]. In fact, ionocytes are already detected in the forming gills even before secondary lamellae, the final structure for gas exchange, start differentiating [87]. The evolution of the expression profile of CEACAMz1 that we report during development is fully consistent with these data and agrees with an expression in epidermal ionocytes. Indeed, we first observe a cutaneous expression of CEACAMz1, from 1 dpf stage. Branchial expression appears at 3–4 dpf and becomes more and more preponderant from this stage, until detectable skin expression totally disappears after 21 dpf, i.e. when gills have reached their full maturity.

Zebrafish epidermal ionocytes derive from non-neuronal ectoderm [88]. They share a common progenitor with keratinocytes and possibly with mucous cells [88–90]. The different categories of epidermal cells start to specify after gastrulation. This process is controlled by Delta-Notch signaling and by Forkhead-box transcription factor class I (*foxi3a*, *foxi3b*)-dependent pathways [88,90]. From 11-somite stage, the five subtypes of ionocytes undergo differentiation from the ionocyte progenitor. Although this process is not yet fully understood, ionocyte differentiation seem to occur again under the control of *foxi3* and Delta-Notch signaling [88,90,91]. HR cell specification is also dependent on transcription factor *gcm2* (glial cell

missing 2) [92]. *Atp6v1a*, the gene coding for the A subunit of $H^+$-ATPase, is turned on at 15-somite stage and *ca2a* (carbonic anhydrase 2a) appears after 18-somite stage [88,90]. The detection of these two robust markers of HR cells indicate that differentiated HR ionocytes start appearing sometime between 16 and 18 hpf. Finally, at 24 hpf, $Na^+/K^+$-ATPase and $H^+$-ATPase proteins can be weakly detected using immunocytochemistry [88,90]. The differentiation process is therefore fully completed by the end of the first day. Based on RT-PCR experiments and analysis of our ceacamz1 transgenic reporter line, we concluded that ceacamz1 expression is turned on sometime between 14 and 16–17 hpf, similarly to other HR cell markers. This suggests that *ceacamz1* is not expressed in the ionocyte progenitor nor in earlier precursor cells, but is instead only detected in fully differentiated HR cells. In consequence, we propose that *ceacamz1* may serve as a novel marker of mature HR ionocytes on zebrafish epidermis.

In various fish species, gill ionocytes have been found to locate mainly along the afferent side of the filaments and intra-lamellar space [82,93,94]. They are tightly associated with the arteriovenous circulation (filament veins and central venous sinus from the inter-lamellar space). This allows them to participate in reversible gill remodeling and modulate the blood-to-water distance according to the ionic conditions imposed by the external environment [94]. In agreement with these observations, we also found our CEACAMz1-expressing HR ionocytes to be localized mainly along the afferent filament arteries at juvenile stage (Fig 8B–8D). No CEACAMZ1-positive cells were observed along the lamellae. Although we could not visualize the central venous sinus (CVS) in our 3D-reconstructions of the gill vasculature, ionocytes expressing CEACAMz1 appeared to align on the lateral side of the afferent filament arteries, from which the lamellar arteries protrude (Fig 8C). This localization suggests that HR ionocytes may also be present in the inter-lamellar space and associate with CVS in zebrafish. The main function of epidermal HR cells resides in acid secretion and $Na^+$ uptake [68,83]. It is comparable to the function of proximal tubular cells in mammalian kidneys [95]. Several studies have allowed to elucidate the molecular mechanisms at play in these processes [73,74,96,97]. Carbon dioxide present as $HCO_3^-$ in the external environment is first dehydrated into $CO_2$ and then taken up on the apical side of HR cells. Inside the cells, it is rehydrated into $HCO_3^-$ by carbonic anhydrase 2a (CA2a). This reaction liberates $H^+$ ions that are in turn secreted outside the cells by apical $H^+$-ATPase and $Na^+/H^+$-exchanger NHE3. This triggers the simultaneous uptake of $Na^+$ ions within the cells. Those are further excreted on the basolateral side of the HR cell membrane through $Na^+/K^+$-ATPase, while trans-epithelial secretion of $HCO_3^-$ is mediated by the basolateral anion exchanger AE1b. As the physiological function of HR cells is tightly connected to $HCO_3^-$ transport, it is not surprising that they concentrate along afferent arteries, which carry deoxygenated blood with high carbon dioxide pressure [82].

From our current data, we cannot tell whether CEACAMz1 may be involved in the ionoregulatory function of HR cells. A direct participation in ion transport seems rather unlikely since the receptor does not possess a transmembrane domain. On the other hand, its Ig domains may promote contacts with neighboring cells through homo- or heterophilic association with other Ig-containing cell surface proteins. Membrane-anchored proteins from the immunoglobulin superfamily are often involved in cell adhesion processes [98]. CEACAMz1 could therefore help shaping the communication between HR cells and adjacent pavement cells or other basal cells, including pillar cells in the inter-lamellar space or other cells from the filament epithelium [94]. Why CEACAMz1 expression would restricts to HR cells in that case remains unclear. But CEACAMz1 can evidently not induce intracellular signaling since it lacks a transmembrane segment. To transmit information within the cells, it therefore needs to act in concert with a co-receptor, as this has been postulated for the human CEACAMs that

are GPI-anchored. One possible explanation for the presence of CEACAMz1 in the sole HR cells may therefore be that its co-receptor is only present in this sub-type of ionocytes.

An alternative function for CEACAMz1 may be to serve as a platform towards the external environment. We clearly observe that CEACAMz1-expressing ionocytes display an apical exposure towards the external medium, at the junction between pavement cells (Fig 5). Our current data do not allow us to conclude as to whether CEACAMz1 expression on the HR cell membrane is apical or basolateral. In case of an apical localization, CEACAMz1 would be in good position to interact with the surrounding environment. Zebrafish epidermis has a more complex cellular composition than human epidermis to insure efficient protection against the more aggressive aquatic environment. In particular, the intermediate layer underlying the surficial network of keratinocytes contains mucous cells that resemble goblet cells from the mammalian intestine and secrete different gel-forming mucins. In that respect, zebrafish epidermis resembles a mucosal epithelium such as the one encountered in mammalian intestine. Thus, although CEACAMz1 is not expressed by the same type of cells as human CEACAMs in the intestine, it lies in the same pathophysiological context as its human counterparts, i.e. it is embedded in an epithelium that is separated from the external environment by a mucus layer. The presence of CEACAMz1 in such an epithelium may therefore parallel the presence of human CEACAMs on the intestinal epithelium. Whether this may reflect a parallelism in physiological function, including possible interaction of CEACAMz1 with zebrafish pathogens, remains to be determined. In conclusion, further studies await to characterize the physiological function of this putative member of the CEACAM receptor family but our data provide evidence that CEACAMz1 constitutes a novel marker of HR ionocytes on the zebrafish epidermis.

## Supporting information

**S1 Fig. Complete DNA sequence of the *ceacamz1* coding region.**
(PDF)

**S2 Fig. Imaging of the transgenic line *Tg(ceacamz1:mCherry-F) ump9Tg* from 10 to 24 somites stages.**
(PDF)

**S1 Raw images.**
(PDF)

## Acknowledgments

We are grateful to Vicky Diakou and Elodie Jublanc for their precious help with the spinning disk microscope and with the IMARIS software. We thank the Montpellier RIO Imaging microscopy platform, member of the national France-BioImaging infrastructure supported by the French National Research Agency (ANR-10-INBS-04, «Investments for the future») for confocal microscopy. We thank Prof. Shigehisa Hirose and Dr. Nobuhiro Nakamura for the gift of the anti-Na$^+$,K$^+$-ATPase and anti-vH$^+$-ATPase antibodies. We are grateful to Dr. Jana Travnickova and Dr. Mai Nguyen-Chi for technical assistance with whole mount *in situ* hybridization and to Panajot Kristofori for help with fish breeding.

## Author Contributions

**Conceptualization:** Georges Lutfalla, Laure Yatime.

**Formal analysis:** Laure Yatime.

**Funding acquisition:** Georges Lutfalla.

**Investigation:** Julien Kowalewski, Théo Paris, Catherine Gonzalez, Etienne Lelièvre, Lina Castaño Valencia, Morgan Boutrois, Camille Augier, Georges Lutfalla, Laure Yatime.

**Methodology:** Georges Lutfalla, Laure Yatime.

**Resources:** Georges Lutfalla, Laure Yatime.

**Supervision:** Laure Yatime.

**Validation:** Laure Yatime.

**Visualization:** Julien Kowalewski, Théo Paris, Laure Yatime.

**Writing – original draft:** Laure Yatime.

**Writing – review & editing:** Laure Yatime.

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
