## [Decision Letter · Decision Letter 0]

25 May 2021

PONE-D-21-13921

Characterization of a member of the CEACAM protein family as a novel marker of proton pump-rich ionocytes on the zebrafish epidermis

PLOS ONE

Dear Dr. YATIME,

Thank you for submitting your manuscript to PLOS ONE. After careful consideration, we feel that it has merit but does not fully meet PLOS ONE’s publication criteria as it currently stands. Therefore, we invite you to submit a revised version of the manuscript that addresses the points raised during the review process.

Please reply each comment raised by the reviewer. In addition, Fig. 3 mentioned in # 1 and 2 of comments should be Fig. 4.

We look forward to receiving your revised manuscript.

Kind regards,

Sheng-Ping Lucinda Hwang, Ph.D.

Academic Editor

PLOS ONE

Journal Requirements:

Reviewers' comments:

Reviewer's Responses to Questions

**Comments to the Author**

1. Is the manuscript technically sound, and do the data support the conclusions?

Reviewer #1: Yes

2. Has the statistical analysis been performed appropriately and rigorously? 

Reviewer #1: Yes

3. Have the authors made all data underlying the findings in their manuscript fully available?

Reviewer #1: Yes

4. Is the manuscript presented in an intelligible fashion and written in standard English?

Reviewer #1: Yes

5. Review Comments to the Author

Reviewer #1: Overall, this is an excellent piece of work. The expression results, including the PCR analysis for tissue distribution and temporal expression of developing stages, in situ hybridization/ immunocytochemistry, as well as the fluorescent images from the transgenic lines, are clear, well presented, and convincing. These results bring us a new marker for HR cells and expand our pieces of knowledge for ionocytes functions and their potential interactions with adjacent cells or the surrounding environment.

The only weakness of this study is the lack of a bit in-depth functional analysis of ceacamz1, such as gain- or loss-of-function and external stimulation experiments (different pH or ion strength, pathogen exposure…). Although the expression data connected ceacamz1 role to the HR cell function, readers still have no idea about its exact physiological role in fish or how it involves in the ionocyte functions.

On the whole, though, this is an interesting and very valuable study. I only provide a few minor comments/questions below to be considered before publication.

1. The quality of Fig. 3 (A-G) needs to be improved. I assume it’s the problem of image acquisition rather than that of ISH staining. It’s better to show a clear image covering the whole body, including the yolk sac part, where abundant ionocytes also distribute.

2. Fig3.: Did the authors performed the ISH of ceacamz1 at 18 hrs or earlier stages? Because the RT-PCR result doesn’t necessarily represent its expression in ionocytes.

3. Did the authors check the expression patterns of other putative ceacam genes? Are there any other family members abundantly expressed in gill? Since some subtypes of ionocytes share the common structural features (intercalated between keratinocyte, direct contact with external environment….), why is the ceacamz1 uniquely expressed in HR cells if its functions are related to cell adhesion or protection against external pathogens?

4. Discussion Line536-547: The authors try to analogize the zebrafish epidermis/ionocytes to the mammalian intestine epithelium/ goblet cells. However, there exists the mucous secreting cell in the gill or skin epithelium. I’m just curious why the mucous secreting in fish epidermis didn’t show the expression of ceacamz1 if I understand the “parallelism” correctly? (if not, please deifine the “parallelism”) or Is the mammalian ceacamz gene also expressed in ion-transporting cells other than the mucous-secreting goblet cells in intestine epithelium?

6. PLOS authors have the option to publish the peer review history of their article (what does this mean?). If published, this will include your full peer review and any attached files.

Reviewer #1: No

---

## [Author Response · Author response to Decision Letter 0]

24 Jun 2021

We thank the reviewer for his/her appreciation of our work and for the very constructive comments which have helped us improve our manuscript. We have taken into consideration all the concerns raised by the reviewer and added new experiments/figures to answer some of the questions. Please find below a detailed, point-by-point response to the reviewer’s comments.

Reviewers' comments:

Reviewer #1: Overall, this is an excellent piece of work. The expression results, including the PCR analysis for tissue distribution and temporal expression of developing stages, in situ hybridization/ immunocytochemistry, as well as the fluorescent images from the transgenic lines, are clear, well presented, and convincing. These results bring us a new marker for HR cells and expand our pieces of knowledge for ionocytes functions and their potential interactions with adjacent cells or the surrounding environment.

The only weakness of this study is the lack of a bit in-depth functional analysis of ceacamz1, such as gain- or loss-of-function and external stimulation experiments (different pH or ion strength, pathogen exposure…). Although the expression data connected ceacamz1 role to the HR cell function, readers still have no idea about its exact physiological role in fish or how it involves in the ionocyte functions.

Answer: we thank the reviewer for his/her appreciation of our work. We agree that functional studies on the physiological role of ceacamz1, in connection with HR-cells, would give more depth to the study. This requires much more work and we could unfortunately not provide such data at this stage of the study, especially due to the work restrictions imposed by the Covid-19 pandemic which slowed down our work quite substantially. We hope to be able to answer these questions in a follow-up study.

On the whole, though, this is an interesting and very valuable study. I only provide a few minor comments/questions below to be considered before publication.

1. The quality of Fig. 3 (A-G) needs to be improved. I assume it’s the problem of image acquisition rather than that of ISH staining. It’s better to show a clear image covering the whole body, including the yolk sac part, where abundant ionocytes also distribute.

Answer: we now provide a new Fig.4 for ISH analysis where images covering the whole body of the embryo/larva are shown. For each developmental stage, images for 2 representative larvae hybridized with antisense RNA probe are shown, followed by a panel displaying a zoom on the region where ISH staining is most visible. As in the former Fig. 4, only 1 representative larva is shown for hybridization with the sense RNA probe, but the image included now covers the whole body of the larva.

2. Fig.3: Did the authors performed the ISH of ceacamz1 at 18 hrs or earlier stages? Because the RT-PCR result doesn’t necessarily represent its expression in ionocytes.

Answer: when performing ISH, we had quite substantial damage of the yolk sac at early larval stages. As a consequence, we could not properly visualize the ISH signal on the yolk region and the images recorded were of insufficient quality for publication. For this reason, we did not attempt extensively to perform ISH at earlier stages than 1 dpf and no ISH data obtained before1 dpf are shown in the manuscript. However, we do agree with the reviewer that it is important to know the distribution profile of ceacamz1 at the very beginning of its expression (i.e. sometimes between 14 and 18 hpf as suggested by our RT-PCR experiments).

To answer the reviewer’s question, we have now added a figure in the supporting information (S2 Fig) where we follow the apparition of the mCherry signal of our Tg(ceacamz1:mCherry-F) reporter line at different developmental stages ranging from 10 to 24 somites (14 to 21 hpf). We recurrently observe that mCherry signal starts being clearly visible at around 16 somites stage (17 hpf) and it is localized on the yolk and yolk extension, as observed later on during development. As mRNA would appear slightly earlier, maybe 1 or 2 hours prior to CEACAMz1 protein, and as it is not detected at 14 hpf, our data suggest that almost from the beginning, of its expression, ceacamz1 localizes in the HR-ionocytes present on the ventral skin of the embryos. We have included an additional paragraph in the manuscript to describe these new findings.

3. Did the authors check the expression patterns of other putative ceacam genes? Are there any other family members abundantly expressed in gill? Since some subtypes of ionocytes share the common structural features (intercalated between keratinocyte, direct contact with external environment….), why is the ceacamz1 uniquely expressed in HR cells if its functions are related to cell adhesion or protection against external pathogens?

Answer: so far, we have not checked for expression of other putative ceacam genes, mainly because there is very little data on them and their categorization in the ceacam family has not been confirmed. But of course, we would very much like to do so in the future, in a follow-up study. 

Almost no expression data are available on putative genes from the ceacam family in zebrafish. In fact, there are only 2 publications in the literature describing the existence of ceacam orthologues in Danio rerio, based solely on genome analysis and sequence comparisons. Data on gene expression for only one other putative ceacam gene are available and suggest that it may be expressed predominantly in the intestine (zgc:198329) but no expression has been reported in the gills and it is still not clear whether this gene belongs to the ceacam family or not.

It is indeed intriguing that ceacamz1 is only expressed in HR-ionocytes if its main function resides in cell adhesion or interaction with the external environment. On the other hand, we cannot rule out that the receptor is involved in other cellular processes that are specific to HR-cells and may relate to ionoregulation or gas exchange. Furthermore, in humans, many ceacam proteins are GPI-anchored, which forces them to use a yet uncharacterized co-receptor for signal transduction. Ceacamz1 is also GPI-anchored. One explanation as to why it is only expressed in HR-cells may also be that its co-receptor is only expressed in HR-cells. We have now added a few lines in the Discussion to address this point.

4. Discussion Line536-547: The authors try to analogize the zebrafish epidermis/ionocytes to the mammalian intestine epithelium/ goblet cells. However, there exists the mucous secreting cell in the gill or skin epithelium. I’m just curious why the mucous secreting in fish epidermis didn’t show the expression of ceacamz1 if I understand the “parallelism” correctly? (if not, please define the “parallelism”) or Is the mammalian ceacam gene also expressed in ion-transporting cells other than the mucous-secreting goblet cells in intestine epithelium?

Answer: in the intestine, human ceacam proteins are expressed predominantly in goblet cells and intestinal epithelial cells. So it is true that it is not a perfect parallelism in terms of cell-type expression since ceacamz1 is not expressed in mucous-secreting cells. What we meant by parallelism was rather a parallelism in terms of patho-physiological context, i.e. that ceacamz1 is expressed in an epithelium covered with a mucus layer that acts as a barrier against external invaders. In that respect, it mimicks perfectly the physiological context encountered by human ceacams. To clarify this point, we have no rephrased the corresponding paragraph in the Discussion section.

---

## [Editor Report · Decision Letter 1]

29 Jun 2021

Characterization of a member of the CEACAM protein family as a novel marker of proton pump-rich ionocytes on the zebrafish epidermis

PONE-D-21-13921R1

Dear Dr. YATIME,

We’re pleased to inform you that your manuscript has been judged scientifically suitable for publication and will be formally accepted for publication once it meets all outstanding technical requirements.

Kind regards,

Sheng-Ping Lucinda Hwang, Ph.D.

Academic Editor

PLOS ONE
---

## [Editor Report · Acceptance letter]

1 Jul 2021

PONE-D-21-13921R1 

Characterization of a member of the CEACAM protein family as a novel marker of proton pump-rich ionocytes on the zebrafish epidermis 

Dear Dr. Yatime:

I'm pleased to inform you that your manuscript has been deemed suitable for publication in PLOS ONE. Congratulations! Your manuscript is now with our production department. 

Kind regards, 

on behalf of

Dr. Sheng-Ping Lucinda Hwang 

Academic Editor

PLOS ONE